# Catalytic Nanomedicine as a Therapeutic Approach to Brain Tumors: Main Hypotheses for Mechanisms of Action

**DOI:** 10.3390/nano13091541

**Published:** 2023-05-04

**Authors:** Tessy López-Goerne, Francisco J. Padilla-Godínez

**Affiliations:** Nanotechnology and Nanomedicine Laboratory, Department of Health Care, Metropolitan Autonomous University—Xochimilco, Mexico City 04960, Mexico; franciscopadilla1997@gmail.com

**Keywords:** catalytic nanomedicine, bionanocatalyst, cancer therapy, nanoparticle, mitochondria

## Abstract

Glioblastoma multiforme (GBM) is the most aggressive primary malignant tumor of the brain. Although there are currently a wide variety of therapeutic approaches focused on tumor elimination, such as radiotherapy, chemotherapy, and tumor field therapy, among others, the main approach involves surgery to remove the GBM. However, since tumor growth occurs in normal brain tissue, complete removal is impossible, and patients end up requiring additional treatments after surgery. In this line, Catalytic Nanomedicine has achieved important advances in developing bionanocatalysts, brain-tissue-biocompatible catalytic nanostructures capable of destabilizing the genetic material of malignant cells, causing their apoptosis. Previous work has demonstrated the efficacy of bionanocatalysts and their selectivity for cancer cells without affecting surrounding healthy tissue cells. The present review provides a detailed description of these nanoparticles and their potential mechanisms of action as antineoplastic agents, covering the most recent research and hypotheses from their incorporation into the tumor bed, internalization via endocytosis, specific chemotaxis by mitochondrial and nuclear genetic material, and activation of programmed cell death. In addition, a case report of a patient with GBM treated with the bionanocatalysts following tumor removal surgery is described. Finally, the gaps in knowledge that must be bridged before the clinical translation of these compounds with such a promising future are detailed.

## 1. Introduction

Primary malignant brain tumors are detected in 5–6 instances per 100,000 people each year, with malignant gliomas accounting for roughly 80% of these occurrences [1,2]. The most prevalent type of primary brain tumors are gliomas, which also include astrocytoma, oligodendrogliomas, and ependymomas [3]. Malignant gliomas are divided into grade III/IV tumors, such as anaplastic astrocytoma, anaplastic oligodendroglioma, anaplastic oligoastrocytoma, and anaplastic ependymomas, and grade III/IV tumors, such as glioblastoma multiforme (GBM), according to the World Health Organization (WHO) [4]. GBM accounts for 80% of all initial malignant CNS tumors, 45.2% of all malignant central nervous system (CNS) tumors, and around 54.4% of all malignant gliomas [5]. The average age of diagnosis is 64 years, and the prevalence is 1.5 times higher in males than in women [6]. Over the past 20 years, the incidence has marginally increased, mostly as a result of advancements in radiologic diagnosis, particularly in older patients [7].

Clinically speaking, GBM patients may have seizures, focal neurologic abnormalities, disorientation, memory loss, or headaches [8]. Magnetic resonance imaging (MRI) and adjunct technologies such as functional MRI, diffusion-weighted imaging, diffusion tensor imaging, dynamic contrast-enhanced MRI, perfusion imaging, proton magnetic resonance spectroscopy, and positron-emission tomography can help with diagnosis and treatment response [9].

Grade III tumors and GBM are handled similarly when it comes to therapy. The standard of treatment for individuals younger than 70 with newly diagnosed GBM is now maximum surgical resection with radiation plus concurrent and adjuvant temozolomide administration [10]. Nevertheless, recurrence appears to be the rule despite regular treatment. Recently, research has focused on figuring out the early molecular pathogenesis of these tumors, including changes in cellular signal transduction pathways, the formation of therapeutic resistance, and approaches to more easily cross the blood–brain barrier (BBB) [11]. Yet, despite these treatments, the illness is still incurable, and the outlook is bad, with a survival range of 6 to 15 months (median 14.6 months) and a mean survival rate of just 3.3% after 2 years and 1.2% at 3 years [12]. Given this scenario, new approaches are urgently needed.

In this regard, supramolecular nanomedicines are causing a paradigm change since they have the potential to diagnose and cure brain cancers [13]. Several nanomedicines are now in clinical use, and many more are making considerable progress in human research. Nanomedicines provide several benefits over conventional formulations, including better tissue selectivity, decreased toxicity, improved drug solubility, and bioavailability [14,15,16]. Nanomedicines can successfully transport drugs over the BBB by fine-tuning their physicochemical properties, including chemistry, size, shape, charge, and surface modifications [17]. In addition, a type of nanomedicine called bionanocatalysts has demonstrated the ability to not only transport chemotherapeutic agents but also to enhance their effect and carry out intrinsic selective cytotoxic activity through the catalytic cleavage of the genetic material of cancer cells, without harming surrounding healthy tissues [18]. This branch of nanomedicine is known as Catalytic Nanomedicine.

Therefore, throughout this review, we thoroughly explain the most promising results regarding these novel bionanocatalysts, which are emerging as potential new therapies for malignant gliomas such as GBM. To better understand the biochemical and molecular basis that must be considered when these treatments are created, we first briefly review our present understanding of brain cancer pathology. The most important developments in the use of bionanocatalysts for the selective elimination of cancer cells are then discussed. Additionally, we describe the main hypotheses regarding the mechanisms of action of these nanomedicines in terms of selectivity, efficacy, and biocompatibility. In support of the above, we include a clinical case of the therapeutic potential of bionanocatalysts against a GBM in a patient. We conclude by providing a succinct overview of the major drawbacks and security issues to take into account when creating nanomedicine-based treatments, particularly in the areas of surface coating and nanotoxicity, because only by addressing these knowledge gaps will research be able to enable further clinical translation. The aforementioned aims to increase our comprehension of the mechanisms underlying the actual perspectives that bionanocatalysts offer for researchers to advance in the search for a cure for GBM.

## 2. Traditional Nanomedicine in Brain Tumors: Enhanced Drug Delivery

Among novel approaches against cancer, traditional nanomedicine has increasingly addressed the challenge of tumor elimination by exploiting their unique properties, such as improved bioavailability of beneficial compounds through controlled pharmacokinetics and pharmacodynamics, blood–brain barrier crossing function, improved distribution to brain tumor sites, and tumor-specific drug activation [19].

To date, the main approaches have focused on enhanced drug delivery. Nanoformulations help to protect chemotherapeutic compounds from chemical and enzymatic degradation in the systemic circulation, in addition to protein binding and reduced uptake by normal tissues (non-reticuloendothelial system), thus increasing accumulation in tumors via passive targeting [20]. There are currently several approved nanodrugs using lipid-based nanotechnology platforms for chemotherapeutic drug delivery. Doxil^TM^ (Bridgewater, NJ, USA) was the first one approved by the FDA in 1995 [21]. Like Doxil^TM^, the liposome-based nanoformulations Caelyx^TM^ (Kenilworth, NJ, USA) and Myocet^TM^ (Castleford, UK) were also approved by the EMA in 1996 and 2000, respectively [22,23]. The three nanodrugs, Doxil^TM^, Caelyx^TM^ (PEGylated), and Myocet^TM^ (non-pegylated), stably encapsulate and retain chemotherapeutic doxorubicin. They were formulated to improve the safety profile of the chemotherapeutic, which is characterized by its high cardiotoxicity [24]. By reducing cardiotoxicity, a higher cumulative dose can be administered in comparison to free doxorubicin [25]. Doxorubicin is an anthracycline antibiotic that is generally believed to interact with DNA via intercalation, inhibiting macromolecular biosynthesis [26]. Doxorubicin stabilizes the topoisomerase II complex after it has broken the DNA strand for replication, preventing the DNA double helix from resealing and halting the replication process. Another reported mechanism of doxorubicin is its ability to generate free radicals that induce DNA and cell membrane damage [27]. Doxil™, Caelyx^TM^, and Myocet^TM^ exhibit prolonged circulation time and reduced volume of distribution, which enhances tumor uptake and extends tumor therapy efficacy [28,29,30]. Doxil^TM^ and Caelyx^TM^ are indicated for the treatment of ovarian cancer, AIDS-associated Kaposi’s sarcoma, and, in combination with bortezomib, for the treatment of multiple myeloma. Caelyx^TM^ and Myocet^TM^ are also indicated for patients with metastatic breast cancer [31,32,33].

DaunoXome^TM^ (Craigavon, UK) is another nanoliposomal preparation that encapsulates daunorubicin, an anthracycline antibiotic with strong antineoplastic activity [34]. The activity of daunorubicin has been attributed mainly to its intercalation between the base pairs of native DNA [35]. It causes DNA damage, such as fragmentation and single-strand breaks. There are two limiting factors in the use of anthracyclines as antitumor agents: chronic or acute cardiotoxicity and spontaneous or acquired resistance [36]. Daunorubicin has a particular affinity for phospholipids and the development of resistance is related to some membrane alterations [37]. DaunoXome^TM^ is indicated as the first line for advanced HIV-associated Kaposi’s sarcoma [38]. Similar to DaunoXome^TM^, another compound is Mepact^TM^ (Cambridge, MA, USA), a liposomal phosphatidylethanolamine muramyl tripeptide that activates monocytes and macrophages [39]. Mepact^TM^ is indicated in children and young adults for the treatment of high-grade resectable nonmetastatic osteosarcoma after complete surgical resection. Furthermore, Ameluz^TM^ (Leverkusen, Germany) is a gel formulation containing INN-5-aminolevulinic acid in a nanoemulsion, enhancing its penetration into the epidermis [40]. The substance is metabolized to protoporphyrin IX and activated by a red light, forming a reactive oxygen species and destroying the target cells. It is indicated for the treatment of mild to moderate actinic keratosis on the face and scalp [41].

Continuing with the list, Marqibo^TM^ (Henderson, NV, USA) is a sphingomyelin and cholesterol-based formulation of vincristine nanoparticles [42]. It is an antineoplastic drug with a broad spectrum of activity against hematological malignancies and childhood sarcomas. It induces neurotoxicity and peripheral neuropathy in a dose-dependent manner [43]. The liposomal carrier component facilitates vincristine loading and retention; extravasation; and slow drug release into the tumor microenvironment, and improves the safety profile of vincristine, reducing its side effects [44]. It is indicated in adults with advanced, relapsed, and refractory diseases. Another compound is Onivyde^TM^ (Cambridge, MA, USA), an irinotecan (DNA topoisomerase I inhibitor) that is encapsulated in a lipid bilayer vesicle, which prolongs circulation time and enhances irinotecan delivery in tumors with compromised vasculature [45]. It is indicated for the treatment of metastatic pancreatic adenocarcinoma in patients who have progressed after gemcitabine-based therapy. Finally, VyxeosTM (Dublin, Ireland) is a liposomal formulation of a fixed combination of daunorubicin (inhibitor of DNA polymerase activity) and Cytarabine (a cell cycle phase-specific antineoplastic agent) [46]. It has a prolonged plasma half-life and accumulates and persists in high concentrations in the bone marrow [47].

As a whole, this type of nanostructure allows the stabilization of drugs with chemotherapeutic properties, which improves their bioavailability and decreases their intrinsic toxicity. However, the nanostructure only serves as a carrier for the drug, without having a therapeutic effect on its own. The other type of approach to solid tumors has sought to exert the chemotherapeutic effect by itself by directly attacking the genetic material of the cancer cells, without harming healthy cells. This new approach is based on the use of bionanocatalysts, nanostructures capable of eliminating tumors by their direct interaction with cancer cells via reducing the activation energy when selectively attacking the bonds, as will be described below [48,49].

## 3. Catalytic Nanomedicine in Brain Tumors

### 3.1. Bionanocatalysts

López-Goerne coined the term “bionanocatalyst” in 2013 [50] to refer to nanostructured materials made of pure or mixed oxide matrices that exhibit enhanced catalytic properties (concerning typical solid cores) and inorganic coating groups that mimic cellular ligands, giving them biocompatibility and affinity. Bionanocatalysts have drawn research teams from all around the world in recent years. These nanomedicines are endowed with inherent biochemical features of significant importance in clinical applications, such as biocompatibility and selectivity, thanks to their surface coating [51]. Bionanocatalysts can be engineered to be innocuous to healthy cells while exerting specific cytotoxicity against pathogenic organisms or damaged cells, making them very well-suited for use in disinfection and cancer treatments.

Unlike traditional nanomedicines, bionanocatalysts’ cytotoxicity is based on their capability to destabilize organic bonds, such as carbon–carbon and carbon–nitrogen, in macromolecules, especially nucleic acids [18]. As will be described in detail, when in contact with DNA or RNA, the bionanocatalysts reduce the activation energy of the system by forming complexes with the structure of the nucleotide chains [52]. Such interaction results in the degradation of the molecule via a sequence of combustion and dephosphorylation reactions that ultimately result in the molecules’ C–C and C–N bonds breaking [53]. With the cell’s genetic material altered, it is inhibited from replicating itself.

In the following sections, we will describe in detail the structure of the bionanocatalysts for cancer therapy (onco-bionanocatalysts) and their relationship with the hypothesized mechanism of action concerning their selectivity, efficacy, and biocompatibility for healthy tissues.

### 3.2. Structure of Onco-Bionanocatalysts

Bionanocatalysts, like traditional inorganic nanocatalysts, have a structure that allows them to carry out selective catalytic processes. However, unlike normal nanocatalysts, bionanocatalysts have a series of additional components that provide them with biocompatibility and specificity towards genetic material exclusively in cancer cells. The oxide core is the main component of any bionanocatalyst (Figure 1). It gives the compound most of its physicochemical and catalytic properties, in addition to serving as a scaffold for coating with transition metals that can be further released or used as active sites in the case of a bionanocatalyst [54,55]. The kind of oxide to choose relies on both the physiological environment in which the bionanocatalysts will function and the intrinsic qualities of the oxide, such as catalytic activity, thermal stability, surface area, and mesoporosity.

The onco-bionanocatalysts base a considerable part of their properties on the nanomechanically structured titanium dioxide (titania) that composes them. This oxide provides the nanoparticle with thermal stability, defined crystalline structure, biocompatibility, and the external surface area [56]. These parameters are crucial to preserve the effect of the bionanocatalysts under whatever conditions they are subjected to. In particular, the intrinsic catalytic ability of nanostructured titania has been extensively studied, as it is necessary to destabilize molecules [57]. This is enhanced due to the high surface areas exhibited by these cores, which translates into larger contact zones and, therefore, reaction zones.

However, not only the catalytic core determines the cytotoxic effect of onco-bionanocatalysts. The stabilization of nanoparticulated transition metals in the oxide lattice’s surface is a bionanocatalyst’s second crucial component. Metal nanoparticle coating improves the bionanocatalysts by producing synergistic effects as a result of these metals’ inherent features [58]. Moreover, it has been shown that coating with these metals boosts their effectiveness, requiring lower amounts of the metal to obtain equivalent outcomes. Platinum (Pt) is the transition metal most frequently utilized in the design of onco-bionanocatalysts given its application as a catalytic converter in the full combustion of unburnt hydrocarbons in exhaust to produce carbon dioxide and water vapor [59,60]. This increases its catalytic activity and allows a 100% metal coating of the core. Depending on the bionanocatalyst’s preparation, high selectivity can be acquired for desired products, as was first demonstrated for the hydrogenation of phenylacetylene [61].

Finally, unlike other nanostructures with catalytic properties (such as those used in petrochemistry), the bionanocatalysts for oncological applications have a surface coating with specific molecules, mainly hydracids. By stabilizing the OH groups in the oxide matrix and creating larger stabilities in the solid with high specific areas, the addition of these molecules enhances the physicochemical structure of the external surface [51,62]. Furthermore, as will be described in the following section, surface coating constitutes a vital step in the synthesis of bionanocatalysts since the production of efficient, selective, and specific molecules depends on the surface coating being designed correctly with certain molecules. Its coating is thought to be directly connected to the mechanism of internalization exclusively into cancerous cells.

### 3.3. Selectivity through Receptor-Recognition

López-Goerne et al. [63] observed that onco-bionanocatalysts internalized into cancer cells through a process of endocytosis. In the assay, the capture of real-time transmission electron micrographs allowed the cell incorporation process to be identified in great detail. Figure 2 summarizes this phenomenon.

As can be seen, the bionanocatalysts (identified by their darker coloration in contrast to the cellular components) are located around the surface of the cell membrane (Figure 2a). The micrograph suggests that there is a binding-type interaction between the nanoparticles and the membrane components, as immediate vesicle formation is observed following pinocytosis-like mechanisms (Figure 2b). A closer view (Figure 2c) shows membrane invagination and incorporation of the bionanocatalysts via pinocytosis. Figure 2d confirms the existence of binding between nanoparticles and membrane components since the bionanocatalysts are exclusively located on the luminal face of the early endosome once internalization is complete. This phenomenon suggests that the bionanocatalysts remain attached to the membrane receptors once the endocytosis process is completed.

Based on the above, a hypothesis of the internalization of bionanocatalysts in cancer cells has been developed. As mentioned above, it is hypothesized that the surface coating of the nanoparticles is responsible for the selectivity of the nanostructures, with the superficial coating molecules being the key to interact with the surface membrane receptors. As for now, three possible types of endocytosis have been proposed for the uptake of the bionanocatalysts: (1) receptor-mediated endocytosis; (2) caveolae-dependent endocytosis; and (3) clathrin-mediated endocytosis.

Research is currently focused on elucidating the type of receptor involved, as well as the signaling pathways activated during the process of endocytosis of bionanocatalysts.

### 3.4. Transport towards Mitochondria

Once in the cytoplasm, the early endosome containing the onco-bionanocatalysts undergoes morphological and biological changes accompanied by vesicle trafficking before their release [64]. According to the maturation paradigm, endosomes throughout the endocytic route are temporary and discrete compartments that go through specific phases as they develop [65]. Endosomal maturation is characterized in this model by four changes: (1) an increase in the number of intraluminal vesicles (Figure 3a); (2) an increase in luminal acidification (Figure 3b); (3) lysosome fusion and Rab protein switching (Figure 3c); and (4) endosome escape (Figure 3d).

Seconds after endocytosis, vesicles fuse with each other or with pre-existing early endosomes following the direction of the small GTPase Rab5 [66]. Early endosomes often originate on the cytoplasm’s periphery, where the intracellular pH is somewhat acidic, allowing the receptor-ligand (bionanocatalysts) to be easily separated. Early endosomes function as crucial sorting stations, delivering dissociated bionanocatalysts to late endosomes and eventually to lysosomes, while recycling back vacated receptors into the cell surface. As previously stated, certain membrane receptors are carried to recycling endosomes with membrane-bound lipids for return to the cell membrane, a process mediated by Rab4 and Rab11 [67,68].

The number of luminal vesicles Increases as endosomes sprout inside the endosomal membrane. Maturation from early endosomes to late endosomes is accompanied by a transition from Rab5 to Rab7 binding [69]. Endosomes travel down to the perinuclear area, which is controlled by interactions with dynein and kinesin, and this is followed by increased intravacuolar acidity. Finally, late endosomes merge with lysosomes, a process focused on cargo and intraluminal vesicle destruction. Nonetheless, given onco-bionanocatalysts’ resistance to acidic conditions due to their surface coating and acidic surface [70,71,72,73,74], degradation does not take place. Instead, rupture of the newly formed endolysosome occurs. The escape of the endosome, as this phenomenon is known, can follow several mechanisms, but the current hypothesis suggests the osmolysis pathway [75]. In this mechanism, onco-bionanocatalysts act as a buffer for pH as protons are pumped into the endolysosome by ATPases. In turn, chloride ions are also pumped to maintain charge balance. As the cell attempts to lower the pH of the endolysosome, an osmotic pressure is created that causes endosomal lysis. Although it is not known whether this process causes total membrane rupture or only pore formation [76], the compelling result is the release of onco-bionanocatalysts into the cytosol. There are likely other factors involved in the ability of these nanoparticles to escape from the endosome, so research continues along these lines.

Upon release into the cytosol, onco-bionanocatalysts travel to the mitochondria, where they interact with mitochondrial DNA. Although the precise mechanism regarding transport from lysed endolysosome to these organelles is not fully elucidated, there is a prominent hypothesis. Catalytic nanoparticles, in general, have been shown to have the ability not only to travel by Brownian motion [77], but also to sense certain structures, such as nucleic acids [78], and to be able to actively transport themselves to them [79]. Although it may seem astonishing that a particle can actively move on its own without any external assistance, there are now several well-established processes, including chemical interactions, that may allow onco-bionanocatalysts to “self-propel”. This mechanism is based on bacterial chemotaxis, in which the bacteria swims on a “random walk” until it detects chemicals to which it reacts, resulting in the whole reorientation of the organism towards the higher chemical gradient [80,81].

Hence, the procedure followed by onco-bionanocatalysts may be closely related to the self-propulsion system of *E. coli*. It has been shown that catalytic nanoparticles with a radius of at least 100 nm can swim by a process called “autophoresis” [82,83]. To this end, the catalytic potential of onco-bionanocatalysts allows them to react chemically with “fuel” molecules that are in the solution. Due to the catalyst coverage, a concentration gradient of the reaction products develops across the particle, which creates a fluid flow near the particle surface [84,85]. According to the conservation of momentum, the particle moves against the direction of the fluid flow, just as a rowboat moves against the direction of the oar strokes. The particle can move with the reactive side facing forward or backward, depending on the individual surface reaction and the surface chemistry of the particle. Normally, within a certain range, as the concentration of fuel molecules increases, so does the activity (velocity) of the particles (chemokinesis) [86]. Because of this, a self-propelled particle will accelerate as it moves towards higher fuel concentrations.

Following this theory, the “fuel” molecule would be hydrogen peroxide (H_2_O_2_). In normal cells, ROS are produced at low levels by NADPH oxidases, and the amount of H_2_O_2_ is controlled by the glutathione system. In tumor cells, in contrast, large levels of ROS near the cytotoxicity threshold are created through the mitochondrial respiratory chain, and the H_2_O_2_ concentration is regulated by catalases [87,88,89]. In turn, the catalytic decomposition of H_2_O_2_ is a well-known disproportionation process involving the simultaneous oxidation and reduction of oxygen, giving rise to water and gaseous oxygen [90]. Notably, this reaction is primarily catalyzed by platinum, one of the metals of choice when designing highly efficient self-propelled nanomachines and micromachines. Thus, onco-bionanocatalysts functionalized with platinum nanoparticles could be taking advantage of their catalytic capacity for the transformation of H_2_O_2_ in increasing concentration gradients towards mitochondria for transport into this organelle. This is depicted in Figure 4a.

However, questions remain regarding a possible cancellation of motion due to Brownian motion itself in this model. Research continues to fully elucidate the transport mechanism as a function of the H_2_O_2_ gradient. Similarly, although evidenced that onco-bionanocatalysts internalize into the inner matrix of the mitochondria (Figure 4b–d), it the pathways followed remain unclear.

### 3.5. Catalytic Bond-Breakage in Nucleic Acids

Upon contact with the genetic material of the mitochondria, the onco-bionanocatalyst acts as a three-way catalyst for the selective reduction of three main types of bonds present in the nucleotides of the macromolecule: carbon–carbon, carbon–nitrogen, and carbon–oxygen [18]. The main agent involved in these reactions is Pt, as described above. The coating with Pt nanoparticles in the catalytic TiO_2_ facilitates the full combustion of unburnt hydrocarbons in the exhaust to produce carbon dioxide, molecular nitrogen, and molecular oxygen (Figure 5) [59]. These reactions catalyzed by Pt include carbon monoxide oxidation, nitrogen oxides’ reduction into N_2_ and O_2_, and the combustion of hydrocarbons [91]:(1)$$2\;CO+O_{2}\to 2\;CO_{2}$$
(2)$$2\;NO_{x}\to N_{2}+x\;O_{2}$$
(3)$$2\;C_{x}H_{2x+2}+\left(3x+1\right)O_{2}\to 2xCO_{2}+2\left(x+1\right)H_{2}O$$

The result of the breaking of these bonds is the formation of innocuous compounds such as CO_2_, N_2_, O_2_, and H_2_O. In turn, the destabilization of the nucleotide bonds results in damage to the genetic sequence, mainly in the form of punctual defects such as depurinations, depyrimidinations, and cytosine deaminations, among others [92]. Such alterations result in genotoxicity associated with genomic instabilities and/or epigenetic alterations that activate the intrinsic DNA repair mechanisms of the cell [93].

### 3.6. Apoptosis-like Death

When onco-bionanocatalysts interact with DNA, the catalytic effect of bond-breaking occurs randomly over the entire structure of the genetic material at a high rate of action [52]. When such damage to the structure of the molecule is produced, repair mechanisms become insufficient to meet the demand. According to the current paradigm, if DNA repair fails, cells will die by activating one of the programmed death pathways, such as apoptosis [94]. In other words, in a given tissue, cells with DNA damage that exceeds their repair abilities are removed from the population. DNA damage often activates extra-cellular death receptors (Fas, CD95, Apo-1) and/or intra-cellular mitochondrial apoptotic pathways [95]. The Fas signaling pathway activates receptors, forms the DIS complex (FADD, Fas-associated protein with a death domain, and procaspase-8 and -10), and activates the caspase cascade, leading to DNA cleavage by CAD (caspase-activated DNAase) and inactivation by proteolysis of intracellular proteins by caspase-3 and -7 [96]. The mitochondrial apoptotic pathway, on the other hand, is based on the regulation of cytochrome c release from mitochondria to the cytoplasm and proapoptotic proteins such as Bax (Bcl-2-associated X protein) and Bak (Bcl-2 homologous antagonist killer) [97]. It is regulated by antiapoptotic proteins and proteins such as Bcl-2 (B-cell lymphoma) and Bcl-XL (large B-cell lymphoma). When cytochrome c is released, apoptosomes containing procaspase-9, Apaf-1 (apoptosis protease activator 1), and cytochrome c are formed. The apoptosome, like the DIS complex, triggers the caspase cascade, which results in protein and DNA inactivation.

Although the exact mechanism of apoptosis following DNA damage by onco-bionanocatalysts is not known, several mechanisms may act at the same time, as has been seen in several studies. The conclusive result, either way, is cell death. This allows for the reduction in size of solid tumors and avoids the survival of potential metastatic cells in the tumor bed. For this reason, approaches to the application of onco-bionanocatalysts in solid tumors have been based mainly on the removal of the tumor by surgery and the application of nanoparticles to the tumor bed.

### 3.7. Biocompatibility of Onco-Bionanocatalysts

The primary concern for the development of any new substance with biological uses is biosafety. Throughout the ages, onco-bionanocatalysts have undergone extensive physicochemical characterization for their catalytic applications in selective bond cleavage [73,98,99,100,101]. Additionally, numerous studies have been conducted to see if these nanostructures have the ability to exert selective toxicity, which means that they can only affect the genetic destabilization of cancer cells and/or pathogenic organisms, without affecting the nearby healthy cells and tissues or inducing a host immune response [102,103,104]. This property is referred to as biocompatibility [105]. In vitro, in vivo, and clinical trials have all been used to study the biocompatibility of onco-bionanocatalysts, with positive outcomes.

The role of the oxide as a cytotoxic chemical buffering agent in healthy cells has been investigated at the cellular level. For instance, Lopez et al. [106] found that by limiting the drug’s internalization into healthy cells, the nanostructured TiO_2_ matrix decreased the phototoxic impact of zinc phthalocyanine by up to 80% compared with the pure substance. Similarly, the functionalized matrix of onco-bionanocatalysts without transition metal is harmless to cells, as observed in various cell viability assays [56,107]. Remarkably, relative harmlessness is also seen when the onco-bionanocatalyst is functionalized with transition metals, since various studies have demonstrated a minor drop in the cell survival of healthy lines when exposed to large concentrations of onco-bionanocatalysts [102,108].

Additionally, it has been demonstrated in both animal and human clinical trials that the incorporation of bionanocatalysts in tissues as diverse as the brain, liver, skin, and lung, among others, has no adverse effects on those tissues or the organism as a whole, supporting what has been seen in vitro for healthy cell lines [63,104,109,110,111,112,113,114]. Interestingly, the surface coating and inherent biocompatibility of the nanostructured matrix of bionanocatalysts appear to be closely connected [115]. The biosafety of bionanocatalysts is supported by these findings; nevertheless, additional research on more cell lines and tissues is required to establish their selective toxicity and biocompatibility.

## 4. Clinical Case of GBM Treated with Onco-Bionanocatalysts

Based on the results obtained both in vitro and in vivo, numerous tests have been carried out on patients with different tumor formations, with positive results. Special attention has been paid to solid brain tumors due to their high incidence and poor prognosis. In the following, to exemplify the selective chemotherapeutic power of onco-bionanocatalysts, we present a case study of an adult patient with glioblastoma multiforme.

### 4.1. Clinical Story

A 53-year-old man presented to the local emergency department after experiencing a generalized seizure lasting more than one minute; he had a right-sided weakness. In retrospect, the patient had noticed more subjective “clumsiness” in the hand he could move one month before the last seizure, but did not communicate this to his family or physician. An MRI was obtained along with an EEG. The patient complained of constant level nine headaches, seizures, and subacute progressive neurological deficits. The patient had had antiepileptic treatment and two surgeries for GBM tumor remotion before this visit. After examination, a Karnofsky Performance Score of 90 was assigned, with subtle right-sided pronator drift and hemiplegia. Right-sided foot movements were his only findings on physical examination. Dexamethasone was started since he had symptoms of elevated intracranial pressure, headache, and papilledema.

The typical MRI appearance of glioblastoma is a mass lesion, often iso- to hyperattenuating (bright) compared with normal gray matter, with surrounding hypoattenuation by infiltrating tumor and vasogenic edema [116]. Classically, contrast-enhanced MRI reveals a centrally necrotic enhancing mass. Since vascular proliferation is a hallmark of glioblastoma, intra-tumoral hemorrhage is common and can be visualized on MRI, although it is more frequently identified as microbleeds on susceptibility-weighted imaging (SWI) MRI [117]. Calcification is uncommon in glioblastoma, but can occasionally be seen [118]. On MRI, almost all glioblastomas are gadolinium-enhanced [119]. In the case of this patient, MRI revealed the presence of a tumor in the right hemisphere (Figure 6).

### 4.2. Treatment

The goals of surgery are tissue diagnosis, including molecular analysis of the tumor, as well as immunohistochemistry for symptom freedom and improved tumor control [120]. Tissue diagnosis is the standard for truly inaccessible tumors (such as GBM). Surgery for high-grade gliomas is generally associated with relatively low rates of major complications [121]: perioperative mortality was reported as 1.5% [122]. In summary, surgery remains the first and very important treatment modality for newly diagnosed glioblastomas. Its efficacy in optimizing overall survival is related to the extent of resection, and its safety depends on several intraoperative adjuncts that allow precise localization of the tumor as well as eloquent cortical areas [123].

Given the patient’s clinical history, a third surgery was recommended for tumor resection, with the difference that, in this surgery, the tumor bed was infiltrated with 2 g of onco-bionanocatalysts during craniotomy using a neuronavigator (Figure 7).

### 4.3. Histology Analysis

The histologic diagnosis, in this case, was a WHO grade IV astrocytoma (glioblastoma multiforme). It was an infiltrating astrocytoma showing areas of high cellularity and vigorous mitotic activity with necrosis and microvascular proliferation (Figure 8). WHO diagnostic criteria include the presence of cytologic atypia, mitotic activity, microvascular proliferation, and/or tumor necrosis [124]. Briefly, an infiltrating astrocytoma exhibiting cytologic atypia alone, including elongated, irregular, hyperchromatic nuclei, is considered WHO grade IV (high-grade astrocytoma). The presence of increased cellularity, nuclear atypia, and mitotic activity warrants a WHO designation (anaplastic astrocytoma). Endothelial cells were found mixed with smooth muscle cells/pericytes via hematoxylin-eosin histology. Necrosis surrounded by tumor cells is characteristic of GBM [125]. The classification of astrocytoma is based on the highest histologic classification. Because infiltrating astrocytomas can have considerable regional heterogeneity, especially towards their infiltrating edge into the surrounding parenchyma, it is important to assess whether a biopsy specimen is representative of the entire tumor by correlating histologic, clinical, and radiologic findings [126].

### 4.4. Results and Prognosis

After surgery, the patient exhibited right-sided weakness but then improved. The patient was put in physical therapy for the next few weeks. He had a very favorable improvement, with no pain or secondary sequelae due to the infiltration of onco-bionanocatalysts.

The patient was kept under observation for the next 4 months. No other organ involvement or secondary effects were observed, suggesting biocompatibility of the onco-bionanocatalysts only for tumor tissues. At 4 months a new MRI was performed (not shown) where no recurrence was observed. The patient presented a survival of 5 years after the application of the treatment.

### 4.5. Other Evidences in Solid Tumors

The above study evidences the efficacy of onco-bionanocatalysts to eliminate remaining cancerous tissues following tumor removal, without observing recurrences or side effects associated with the incorporation of the nanoparticles. This work contributes to the extensive research that has been developed so far on the application of onco-bionanocatalysts as a potential treatment for solid brain tumors. In this line, López-Goerne et al. (2020) [63] also demonstrated the efficacy of these nanostructures in the elimination of a pediatric ependymoma tumor, where the treatment allowed survival of the patient without side effects after tumor resection and the application of the onco-bionanocatalysts in the tumor bed. Another solid tumor treated with Catalytic Nanomedicine with promising results is cellular hepatocarcinoma. The research group has demonstrated tumor size reduction without the occurrence of side effects [110,112], suggesting an intrinsic biocompatibility of the nanostructures. Notably, these and other in vivo studies corroborate the results previously evidenced in vitro concerning the biocompatibility and selectivity of onco-bionanocatalysts [102]. In this study, their cytotoxic capacity against cervical-uterine and prostate cancer lines was evidenced, without affecting a normal fibroblast line. Taken together, this evidence supports biosafety in the use of these nanostructures.

## 5. Perspectives and Challenges

Although the usefulness of onco-bionanocatalysts has been shown both in GBM and other types of tumors, the specific processes behind their activity have not yet been fully elucidated since speculations about some events remain unverified. The biggest unanswered question in anticancer therapies is the specific mechanism that gives onco-bionanocatalysts their selectivity for cancer cells without harming neighboring healthy tissues. Although there are proposals related to protein–ligand interactions that facilitate the recognition and endocytosis of onco-bionanocatalysts, there are currently no studies describing the surface-coating agent responsible for this recognition or on the type of membrane receptor particular to cancer cells whose interaction with the ligand results in the uptake of the NPs. A multidisciplinary investigation is now being conducted to examine this interaction, identify the receptors implicated, and determine the endocytosis pathways that are activated as a result of the contact. To clarify the range of effects that these onco-bionanocatalysts can have (especially concerning the most aggressive and common cancers), as well as confirm their biocompatibility with the most crucial healthy tissues, it is also necessary to increase the number of studies in various cell lines, both cancerous and healthy.

It is also necessary to further investigate new routes of administration, biodistribution tempering, and the passage of onco-bionanocatalysts through biological barriers, such as the blood–brain barrier. In the present study, the in situ administration of the NPs allows them to have a 100% biodistribution in the tumor bed, which allows a reduction in the concentration required in the application. However, in the case of brain tumors, infiltration of the tumor bed requires quite aggressive routes, such as surgery. In these routes, the transport mechanism hypothesis is based on the enhanced permeability and retention effect, which describes an increased concentration of particles in the nanoscale (such as NPs and liposomes) in the tumor as compared with nearby healthy tissues [127]. This hypothesis proposes that the bioavailability of onco-bionanocatalysts in the tumor might be linked to the tumor microenvironment, which would promote their absorption in the tumor surroundings. However, in these alternatives, the different transport steps before the localization of the onco-bionanocatalysts in the tumor hinder the permanence and interaction with the target site, leading to the use of high concentrations of the NPs [128]. Furthermore, there is a chance of lung, liver, and kidney accumulation even when NPs are regulated appropriately to lengthen the retention duration and half-life, and even if the biocompatibility of onco-bionanocatalysts has been proven. For instance, lung deposition of other types of NPs with inflammatory, oxidative, and cytotoxic effects has been observed [129]. Therefore, research is needed to optimize the bioavailability of onco-bionanocatalysts administered through systemic pathways. In this sense, the above could be improved by surface-coating the nanostructures according to tumor type and location [130] so that their systemic administration can be optimized, and the concentrations required for the necessary concentration to reach the target site can be reduced. Likewise, the ability of onco-bionanocatalysts to cross the blood–brain barrier is unknown so far, so before the use of systemic routes as methods of administration of onco-bionanocatalysts for brain tumors, it is necessary to study this phenomenon and, failing that, to venture into the surface-coating of NPs for their blood–brain transport.

Finally, scale-up synthesis, equal optimization, and performance projections are onco-bionanocatalysts’ technological hurdles. They are extremely important in ensuring these NPs’ clinical success. The onco-bionanocatalysts utilized in in vivo and in vitro investigations are often manufactured in small batches, and scale-up for enormous amounts is frequently not practical due to equipment and other factors. The best lead clinical prospects in animal models are not always systematically conceived and optimized. To get around this, we can employ specific techniques that can evaluate a variety of nanoformulations and pick one optimum formulation through careful iteration [131,132,133]. Such impacts, meanwhile, should not be immediately added to human testing. It is challenging to predict the effectiveness and performance of NPs, and it is impossible to reproduce in vivo findings in human trials. Experimental data and theoretical or computational modeling can be used to create an environment and tissue that mimic physiological conditions.

## 6. Conclusions

Onco-bionanocatalysts are catalytic nanostructures capable of selectively breaking bonds in genetic material, allowing its destabilization. Such damage results in the activation of cell death through apoptosis. Due to their surface-coating, these nanoparticles can identify and internalize cancer cells without interacting with healthy cells, a property that allows them to serve as a potential new treatment for solid tumors without altering the surrounding tissues and, therefore, without associated side effects. In the present work, the main hypotheses behind the selective surface-coating of these onco-bionanocatalysts were presented. Although there are still gaps in knowledge concerning their mechanisms of action, their effectiveness has been demonstrated in countless studies, as evidenced in the treated glioblastoma multiforme patient. The application of the treatment improved the patient’s survival, without generating side effects and preventing tumor recurrence. The research will continue to resolve the key questions that remain to be answered, as described in the Perspectives and Challenges section. Once the mechanisms of action have been fully elucidated, onco-bionanocatalysts could become a highly efficient and safe alternative for the treatment of localized solid tumors.

## Figures and Tables

**Figure 1 nanomaterials-13-01541-f001:**
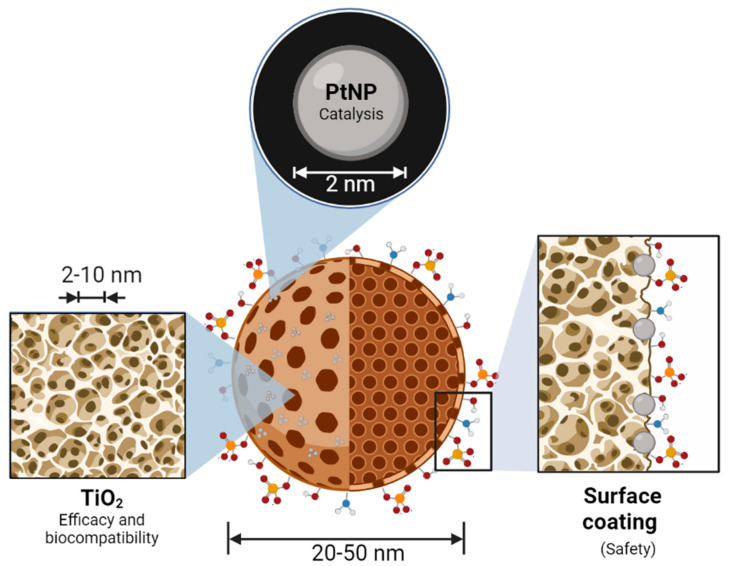
Molecular structure and main components of an onco-bionanocatalyst. The nanostructured oxide matrix provides a platform for the stabilization of transition metal nanoparticles for synergistic effects. In addition, the matrix serves as a catalyst in its own right, enhancing the effect of the metal. The surface coating of the oxide matrix surface determines the selectivity and biocompatibility of the bionanocatalysts. Figure made in BioRender.com (accessed on 13 April 2023).

**Figure 2 nanomaterials-13-01541-f002:**
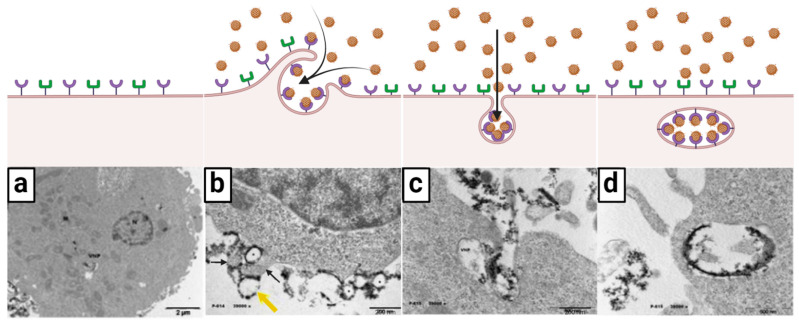
Ligand–receptor-mediated endocytosis: selective internalization mechanism of onco-bionanocatalysts. (**a**) Generally, tumors cells exhibit different types of receptors in the surface of cell membrane. (**b**) When in the presence of onco-bionanocatalysts, certain receptors interact with the superficial coating in the surface of the NPs (the ligands) and carry out a pinocytosis-like process (yellow and black arrows). (**c**) Invagination of the membrane allows for onco-bionanocatalyst internalization. (**d**) As the ligand–receptor interaction remains, onco-bionanocatalysts are observed in the lumen face of the early endosome where the receptors are located. Micrographs from personal archives. Figure made in BioRender.com (accessed on 13 April 2023).

**Figure 3 nanomaterials-13-01541-f003:**
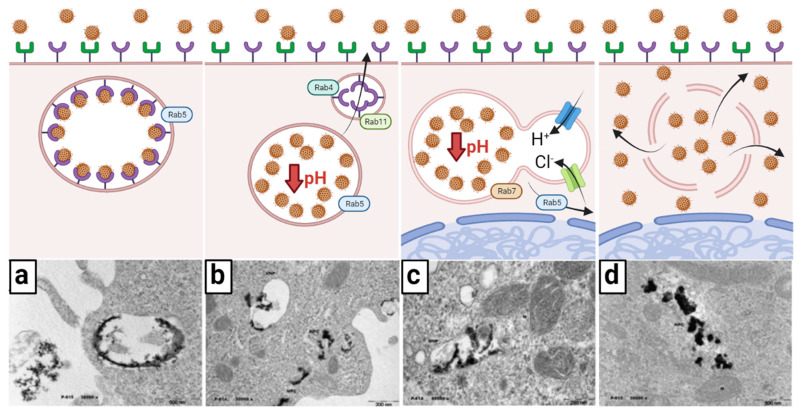
Early transport and endosome escape of onco-bionanocatalysts. (**a**) After endocytosis, an early endosome is made; Rab5 determines the movement of the endosome. (**b**) pH decrease dissociates ligand–receptor interactions and receptors are recycled into the cell surface by Rab4 and Rab11. (**c**) The endosome travels towards the perinuclear region and fuses with lysosomes; pH increases via H^+^ pumping by ATPases; Cl^−^ are also pumped to balance the charge, creating an osmotic pressure. (**d**) The endosome bursts due to excessive osmotic pressure, releasing the onco-bionanocatalysts. Micrographs from personal archives. Figure made in BioRender.com (accessed on 13 April 2023).

**Figure 4 nanomaterials-13-01541-f004:**
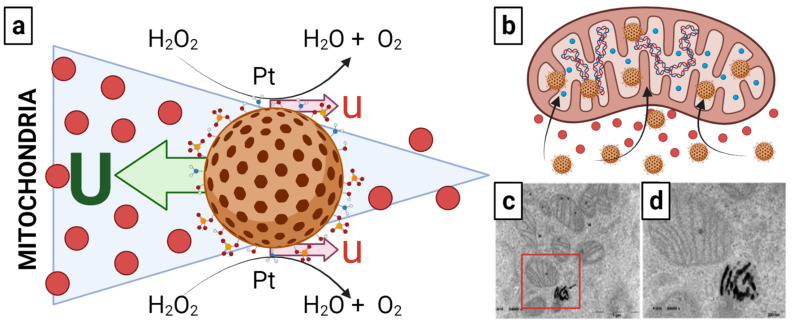
(**a**) Possible autophoresis mechanism followed by onco-bionanocatalysts to travel towards the mitochondria and (**b**–**d**) mitochondria internalization. The slip flows near the surface of the colloid with velocity **u** (a product of catalytic conversion of H_2_O_2_ into H_2_O and O_2_), leading to particle motion with translational velocity **U** in the opposite direction to the respective slip flows because the total force in the system must be balanced. Once in interaction with the mitochondria, the onco-bionanocatalysts internalize into the matrix. Micrograph in (**d**) is an amplification of the region inside the red square in micrograph in (**c**). Micrographs from personal archives. Figure made in BioRender.com (accessed on 13 April 2023).

**Figure 5 nanomaterials-13-01541-f005:**
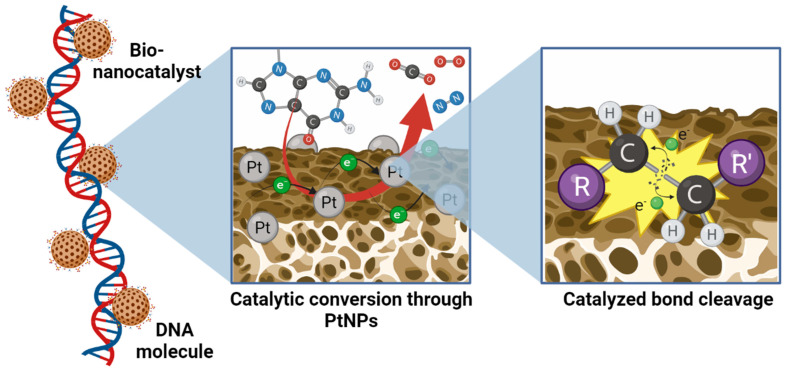
Catalytic bond cleavage of atomic bonds in nucleotides by bionanocatalysts. Pt nanoparticles dispersed and potentiated by the catalytic TiO_2_ matrix selectively carry combustion reactions to break carbon–carbon, carbon–nitrogen, and carbon–oxygen bonds, liberating as sub-products carbon dioxide, molecular oxygen, molecular nitrogen, and water. Figure made in BioRender.com (accessed on 13 April 2023).

**Figure 6 nanomaterials-13-01541-f006:**
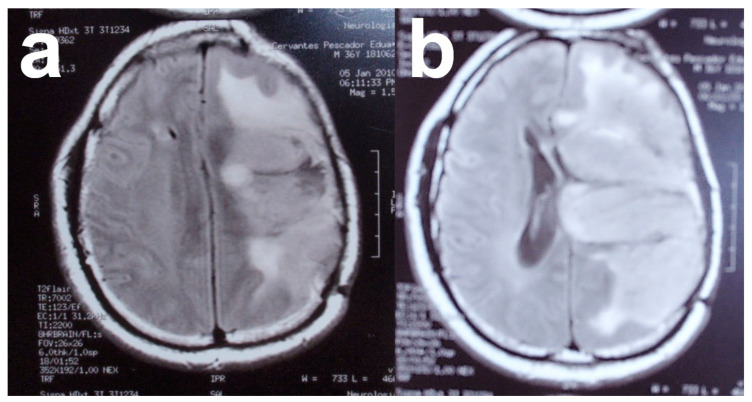
(**a**,**b**) Magnetic resonance images of patient with solid glioblastoma multiforme tumor. Images from personal archive.

**Figure 7 nanomaterials-13-01541-f007:**
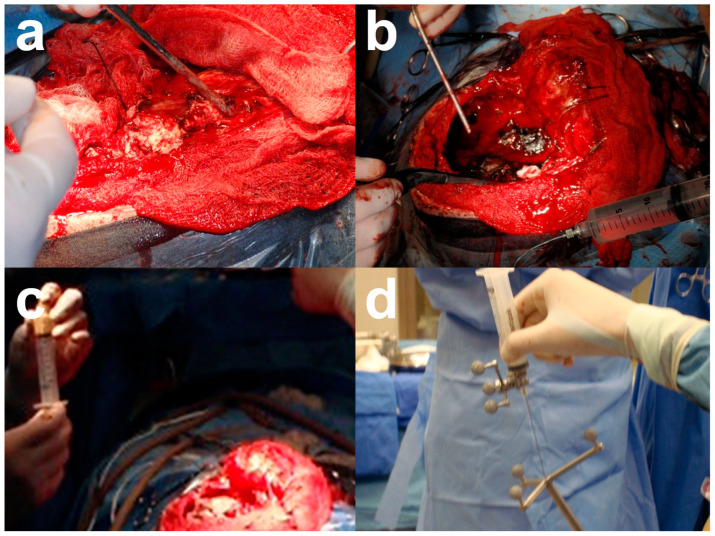
Craniotomy performed in the patient and onco-bionanocatalyst implantation. (**a**) Craniotomy and (**b**) tumor exposure. (**c**) Preparation of onco-bionanocatalysts in distilled water and (**d**) infiltration using a neuronavigator. Images from personal archive.

**Figure 8 nanomaterials-13-01541-f008:**
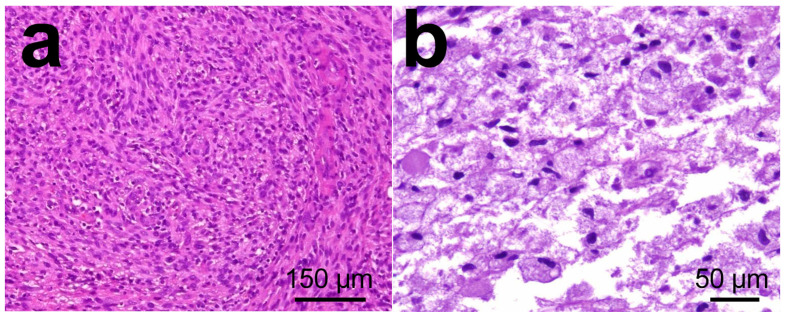
Histology images in glioblastoma multiforme tumors from patient. (**a**) Necrotic area in tumor surrounded by pseudo palisading tumor cells. (**b**) High power magnification of microvascular proliferation. Images from personal archive.

## Data Availability

All data generated and analyzed during the current study are available from the corresponding author on reasonable request.

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
