# Peer review of "Catalytic Nanomedicine as a Therapeutic Approach to Brain Tumors: Main Hypotheses for Mechanisms of Action"

_nanomaterials, 2023, doi:10.3390/nano13091541_

Round 1

Reviewer 1 Report

This paper provided some information about catalytic nanomedicine for the treatment of glioblastoma multiforme. However, several issues should be addressed.

1.     The Section 2 is not necessary since DaunoXomeTM  and MarqiboTM are not specific for glioblastoma multiforme and traditional nanomedicine is more than these two drugs.

2.     The structure of onco-bionanocatalysts should be illustrated before the action mechanism. So Section 3.2 Structure of onco-bionanocatalysts can be moved to be Section 2.

3.     The content about onco-bionanocatalysts only involves PtNP and titanium dioxide. Other onco-bionanocatalysts are suggested to be described and discussed.

4.     More experimental results on cells and animals are needed.

Minor editing of English language required

Reviewer 2 Report

Journal: Nanomaterials

Manuscript ID: nanomaterials-2350302
Type of manuscript: Mini review
Title:  Catalytic Nanomedicine as a Therapeutic Approach to Brain Tumors: Main Hypotheses for Mechanism of Action

Authors: Tessy López-Goerne, Francisco J. Padilla-Godínez

In this manuscript submitted as Mini Review to the Nanomaterials, the authors present and discuss with help of some literature data interesting results obtained by their research group (I assume that the authors belong to the group of T. Lopez). This is why I would recommend to change the paper category, from Mini review to Communication. In the current form is not suitable to be published as Review, since it is obvious that the authors express some ideas, based on their own experience (or experience of their larger group) sustained by some punctual references from the literature data. For a Review, the method must be indicated, telling how many and which data bases were searched, what key words were used, how many articles/reviews were found on the topic(s) of interest, criteria of inclusion/exclusion etc.

Otherwise, the paper is a very interesting one, and the data presented and discussed are of actuality (reflected by the many new references used). The topic of this manuscript is relevant to the field of the Nanomaterials journal and fits with the scope of this journal. The manuscript is well written, it is comprehensive but still concise, and the author’s style captivates attention of the reader. The language is clear and correct, but there are some scientific issues (in text and figures) that must be solved prior to publication. Also, a few corrections are required. Therefore, I would recommend this manuscript for publication in the Nanomaterials journal after a minor revision addressing the following 4 aspects.

1. The most important aspect concerns the type of the paper as mentioned above. If the authors insist to publish the manuscript as Review, they have to include a section with Methodology, describing how they performed the documentation and other aspects suggested above. They could also add into the title „based on the experience of our group” or something similar.

2. Some scientific aspects to be corrected:

- Line 237: please rephrase, since pinocytosis is a type of endocytosis.

- In lines 266-267, and 276 the authors correctly describe the processes inside the endosomes, but they have to correct into the figure 3 the upper panels of b and c, where the arrows show increase of pH (which is not correct, acidification means a higher amount of protons, corresponding to a lower pH). The same comment for the legend of Figure 3b, and 3c (please change „increase” to „decrease”).

- Please remove the paragraph with the forced comparison with bacterial chemotaxis.

- Please add „Pt” next to the curved arrows in Figure 4, or also add „cytosolic catalase” if this enzyme also participates in the conversion of H2O2. Also in the same figure, please replace  according to the legend, or change „U” to „V” in the legend (for consistency).

- It would be helpful to add references to support the transport of onco-bionanocatalysts inside the mitochondria.

- In the legend of Figure 5, why „water vapour”? are the reactions so exergonic that the resulted water evaporates?

- In line 390, it is not clear what articulation is involved in the process.

- In lines 430-433 that important statement has to be supported with references.

- Figure 6 is not presented in the text.

- For Figures 6-8, the authors have to indicate the source. If the images were taken from the literature data, the mention „reproduce with permission” has to be added; If these images were not previously published and belong to their research group, please mention „from personal archive”.

- In line 541, please rephrase, since NPs and liposomes are not „tiny molecules”.

- In line 575, please rephrase „programmed cell death of the cell”.  

3. The authors should perform the following corrections into the text (or rephrase):

- Line 29: plural of person is people

- Line 30: please use newer, updated references for [1,2], since one was published 17 years ago and the other one 13 years ago (or rephrase).

- Line 35: reference [4] is not an official source of WHO. Please change it or rephrase.

- Please rephrase lines 36-37. The current formulation is ambiguous.

- Lines 39-40: please use a newer, updated reference for [7] since this paper was published 16 years ago and cannot sustain a incidence of „the past 20 years”.

- In line 91, reference [19] is actually [13] – the same paper is cited twice. And from here, the authors must change all the numbers (or insert another reference instead of 19 - Quader et al, 2022).

- Line 97: please rephrase.

- Line 150: please change „allow” to „allows” for accord.

- Line 236: please change „Figura” to „Figure”.

- Line 261: please rephrase for accord between „endosome” and „undergo”.

- Line 385: please change „observed” to „produced”.

4. References list:

- Please correct reference 40.

- Please remove reference 57, or replace it with another, more scientific one.

- Please format authors of reference 78.

- Reference 79 seems to be incomplete.

- In reference 90, the title is incomplete.

English is good. Only few minor corrections are required.

Reviewer 3 Report

The authors provided an overview of the primary hypotheses regarding the therapeutic mechanism of catalytic nanodrugs on brain tumors. The subject matter is compelling, particularly the investigation of the catalytic mechanisms of nanomedicines. Nevertheless, there are several noteworthy issues that the authors must address before this manuscript can be accepted.

1. In sections 2 and 3, the authors discuss "Traditional Nanomedicine in Brain Tumors: Enhanced Drug Delivery" and "Catalytic Nanomedicine in Brain Tumors" respectively. It would be beneficial to provide specific examples to support the advantages of catalytic nanodrugs in treating brain tumors.

2. While section 4 presents a clinical case of GBM treated with onco-bionanocatalysts, citing only one case may not be entirely persuasive. It would be helpful to include additional examples to strengthen the case studies presented in this section.

3. Section 3, "Catalytic Nanomedicine in Brain Tumors," includes a discussion of the fundamental structure of nanocatalysts. It may be beneficial to follow the approach taken in the "Traditional Nanomedicine in Brain Tumors: Enhanced Drug Delivery" section by introducing several nanocatalysts that are commonly used in treating brain tumors.

4. Section 3.2 "Tumor-structure of biological nanocatalysts", the common basic characteristics of nanocatalysts were not explained. please add these discussions.

5. The authors write:Following this theory, the "fuel" molecule would be hydrogen peroxide (H2O2). In normal cells, ROS are produced at low levels by NADPH oxidases, the H2O2 concentration is regulated by catalases [85–87]. However, the literature cited in this section is not particularly recent, and it is noted that ROS production is influenced by hydrogen peroxide concentrations. Currently, many catalysts are combined with self-supplied hydrogen peroxide materials to form new nanomedicines for tumor treatment. Could this review mention this point?

6. With numerous reviews available on the topic of nanomedicine and tumors, what sets this particular review apart in terms of its key contributions or unique insights? Some examples of other relevant reviews include: https://doi.org/10.1016/j.jconrel.2014.12.030ï¼›https://doi.org/10.1155/2019/6313242.

  • Some formats are incorrect. Please modify them.
  •  

Round 2

Reviewer 1 Report

It can be accepted for publication.